# Effect of High Pressure on the Properties of Chocolate Fillings during Long-Term Storage

**DOI:** 10.3390/foods11070970

**Published:** 2022-03-27

**Authors:** António Panda, Patrícia Coelho, Nuno B. Alvarenga, João Lita da Silva, Célia Lampreia, Maria Teresa Santos, Carlos A. Pinto, Renata A. Amaral, Jorge A. Saraiva, João Dias

**Affiliations:** 1Faculdade de Ciências e Tecnologia, Universidade Nova de Lisboa, 2829-516 Monte da Caparica, Portugal; ap.neto@campus.fct.unl.pt (A.P.); jfls@fct.unl.pt (J.L.d.S.); 2Instituto Politécnico de Beja, Rua Pedro Soares, 7800-295 Beja, Portugal; patricia.lage@ipbeja.pt (P.C.); celia.lampreia@ipbeja.pt (C.L.); t.santos@ipbeja.pt (M.T.S.); 3UTI, Instituto Nacional de Investigação Agrária e Veterinária IP, Quinta do Marquês, 2780-157 Oeiras, Portugal; nuno.alvarenga@iniav.pt; 4GeoBioTec Research Institute, Campus da Caparica, Universidade Nova de Lisboa, 2829-516 Monte da Caparica, Portugal; 5LAQV-REQUIMTE, Department of Chemistry, University of Aveiro, 3810-193 Aveiro, Portugal; carlospinto@ua.pt (C.A.P.); renata.amaral@ua.pt (R.A.A.); jorgesaraiva@ua.pt (J.A.S.)

**Keywords:** chocolate fillings, high pressure processing, rheological behaviour, shelf life, microbiology

## Abstract

The aim of this study was to evaluate the impact of high-pressure processing (HPP) on the long-term storage of chocolate fillings at room temperature, compared with conventional storage at lower temperatures. Dark chocolate fillings were treated at different pressure levels, holding times and stored at 20 °C for 12 months. Unprocessed batches were stored at 4 °C and at −12 °C. Moisture, water activity (aw), pH, colour, G′_1Hz_ and indigenous microflora were measured at 2, 4, 6, 8, 10 and 12 months of storage. Results showed that 600 MPa/20 min processing was the most effective controlling mesophilic group, presenting 3.8 log cfu/g after 12 months of storage, and inactivating moulds and yeasts after HPP treatment. Colour was affected by storage, including a reduction in the L* parameter in all conditions to final values between 37.8 and 39.3, while the a* parameter increased during storage time at −12 °C and 4 °C to final values of around 12, and parameter b* decreased at storage temperature 20 °C to 5.3. Storage time affected the rheological behaviour of HPP-treated samples, increasing G′_1Hz_ from the 2nd to 12th month of storage time to the final values between 1603 kPa and 2139 kPa. Moisture, aw and pH were not affected by HPP treatment nor storage time.

## 1. Introduction

Chocolate is one of most common confectionery products worldwide and has been studied for many years. Chocolate is a suspension of cocoa mass and sugar in the cocoa butter matrix, solid at room temperature that melts at oral temperature (37 °C) creating a smooth sensation [1]. However, the consumption of chocolate can include several types, other than plain chocolate bars, such as desserts, drinks, confectionery, among others [2]. “Ganache” is one of the most common chocolate preparations, consisting of a mix of chocolate and cream, where other ingredients can also be included. This preparation can be used as a glaze, a filling for pastries or for chocolate bonbons [3,4]. Structurally, ganache is a complex multiphase system and consists of an emulsion where dispersed droplets of one phase are dispersed into another phase, namely oil-in-water (O/W) or water-in-oil (W/O) emulsion [5], depending on a set of parameters such as the production process, type of emulsifier or proportion between ingredients.

There is not a standard preparation for ganache; however, the percentage of cream in a recipe may range from 9 to 41% *w*/*w* [4], and most ganache formulations contain high levels of water from the ingredients, such as cream or fruit purée, providing a pleasant and smooth texture. However, so much water together with a high content of sugar and proteins creates a good culture media for the development of numerous microorganisms such as fungi, yeasts and pathogenic bacteria such as *Salmonella* and *Listeria* spp. [4], with this being the main reason for the reduction in shelf life to just 3–4 months [6]. Additionally, the high amount of nutrients, especially fat, protects these microorganisms from treatments used in conservation processes, thus making chocolate a matrix in which microorganisms are also protected when they cross the digestive tract, resisting acidity in the stomach. Thus, even the presence of small amounts of microorganisms in this type of matrix can be sufficient to cause disease, which has happened in some outbreaks of foodborne infections due to the consumption of chocolate contaminated by *Salmonella* spp. [7]. This situation represents a challenge to the chocolate industry, where the most followed strategies to extend shelf life include the thermal treatment of cream [6], reduction in water activity or pH, use of ethanol, use of preservatives [8] or creating a balance between the humidity of the ganache and the environment [4]. Additionally, refrigeration is an important solution for maintaining food safety high standards and food quality due to the impact on microbiological growth slowdown (e.g., mesophilic growth), physiological processes (e.g., ripening), biochemical reactions (e.g., lipid oxidation) or physical changes (e.g., desiccation). However, using conventional refrigeration systems to remove heat loads especially during long-term storage is energy consuming, causing a significant environmental impact, together with the risk of the leakage of refrigerant into the atmosphere [9]. On the other hand, high pressure processing (HPP) is considered a nonthermal process capable of microbial inactivation by pressure, which is additive-free and with lower power consumption, when compared with conventional refrigeration and freezing [10]. Currently, HPP technology in the food industry has been applied in several areas, including chocolate, dairy products, beverages, among others [10,11,12,13]. Previous investigations conducted in filled chocolates presented satisfactory results regarding the use of HPP to extend shelf life; however, the involved pressures in HPP surpassed the mechanical limits of coverture chocolate, presenting a high rate of broken chocolates and limiting the application in such kinds of products [10].

The aim of this study was to investigate the potential use of different HPP treatments to extend the shelf life of chocolate fillings, compared to standard long-term storage and considering physical, chemical, mechanical and microbiological parameters. The present work is of pertinence considering that there are very few studies covering the use of nonthermal processing technologies, such as HPP, to extend the shelf life of chocolate fillings.

## 2. Materials and Methods

### 2.1. Preparation

Chocolate fillings were produced in a local confectionery (Sugar Bloom/Mestre Cacau, Beja, Portugal) using dark chocolate Apamate 70% (Chocolates El Rey, Venezuela) and cream (33% fat) in a 2:1 proportion. The preparation of chocolate filling started by heating up the cream to 90 °C using a heating plate with a magnetic stirrer. After reaching 90 °C, the chocolate was added using a blender to homogenize the mixture for 2 min (Bimby, Germany). The mixture was allowed to cool down to 30 °C at room temperature. Then, approximately 50 g of filling was piped manually into polyethylene PET-MET/PE bags (Alempack Lda, Portugal) and vacuum packaged at 800 mbar (Audionvac 151H, Weesp, The Netherlands). A similar procedure was adopted for high pressure processed and unprocessed samples.

### 2.2. High Pressure Processing Treatments

HPP treatments were carried out in a 55 L high pressure equipment (Hiperbaric 55, Burgos, Spain) at 400 MPa/10 min, 400 Mpa/20 min, 600 Mpa/10 min and 600 Mpa/20 min. HPP samples are identified in Table 1, according to pressure and exposure time. The HPP equipment includes a pressure vessel of 200 mm inner diameter, 2000 mm length and a maximum operation pressure of 600 Mpa, connected to a refrigeration unit (RMA KH 40 LT, Ferroli, Italy) for temperature control of the inlet water used as the pressurizing fluid at 17 °C.

### 2.3. Storage Conditions

Unprocessed sample Control, NoHPP/−12 and NoHPP/4 (Table 1) were stored at 20 °C, −12 °C and 4 °C, respectively, while samples submitted to HPP were stored at 20 °C. All samples were stored for 12 months in the absence of light, and analyses were performed every two months. Temperature values were recorded every 30 min using a datalogger (Testo 174T, Lenzkirch, Germany). Table 1 summarizes the processing and storage conditions in the present study.

### 2.4. Physical and Chemical Analyses

The moisture content was determined according to the gravimetric method [10]. The water activity values (aw) were measured at 20 ± 1 °C using a portable water activity meter (Hygropalm HP23-A, Rotronic, Bassersdorf, Switzerland), based on the electric hygrometer principle. The pH was evaluated with a penetration electrode at 20 ± 1 °C (Metrohm 691, Herisau, Switzerland). The colour parameters of samples were evaluated using a portable colorimeter (Minolta CR-300, Tokyo, Japan), based on CIE 1976 L*a*b* colour system, where parameter L* is the lightness variable and ranges from 0 (black) to 100 (white), the a* parameter represents the chromaticity from green (negative) to red (positive) and the b* parameter regards the chromaticity from blue (negative) to yellow (positive) [10]. Small amplitude oscillatory measurements were performed using a controlled shear-strain rheometer (Kinexus lab+, Malvern Instruments, Malvern, UK), at 20 ± 1 °C, using a 20 mm serrated parallel plate geometry and 1 mm gap distance. The linear viscoelastic region (LVR) was evaluated by performing a strain sweep (0.001–1000%) at a steady frequency of 1 Hz. The dynamic frequency sweep was conducted applying a steady strain of 0.01%, within the LVR, from 0.01 Hz to 100 Hz. The outputs of each rheological measurement were the mechanical spectrum, i.e., G′ and G″ (in kPa) as a function of a frequency, and the G′ at 1 Hz (in kPa). Physical and chemical analyses were conducted in quintuplicate.

### 2.5. Microbiological Analysis

Samples of chocolate fillings (10 g) were diluted (1:10, in volume) in Ringer solution BR0052G (Oxoid, Hampshire, UK) and homogenized for 2 min in a Stomacher 400 Circulator (Seward, UK). A 1.0 mL aliquot of the homogenate was serially 10-fold diluted and 1 mL aliquots of appropriate dilutions were plated on specific media. Total aerobic mesophiles viable organisms were grown on Plate Count Agar (PCA) CM0325 (Oxoid, Hampshire, UK), incubated at 30 °C for 72 h, *Enterobacteriaceae* on Violet Red Bile Glucose Agar (VRBG) CM0485 (Oxoid, Hampshire, UK), incubated at 37 °C for 24 h, and moulds and yeasts on Rose-Bengal Chloramphenicol Agar Base CM0549 (Oxoid, Hampshire, UK), incubated at 25 °C for 120 h. The limit of detection (LoD) was set to 1 log cfu/g. Microbiological analyses were conducted in duplicate.

### 2.6. Statistical Analysis

The obtained results were subjected to statistical analysis where the significance level was set to 1% and samples having a size less than or equal to two were not considered. In order to use one-way analysis of variance (one-way ANOVA), assumption of normality was previously tested by using Shapiro–Wilk’s test [14]; further, quantile-quantile plots (Q-Q plots) were drawn and analysed, as well as the computation of z-scores for skewness and kurtosis [15]. The homogeneity of variance was tested by performing Levene’s test [16]. When normality assumptions were met, post hoc single-step multiple comparison Tukey’s HSD test [17] was employed to investigate significant differences between mean values, while in the case of heterogeneity of variances, Welch’s test [18] was used, along with Dunnett’s T3 test for determination of all possible pairwise contrasts [19]. All statistical analysis was carried out with IBM SPSS Statistics Version 22 (IBM, Armonk, NY, USA).

## 3. Results

### 3.1. Physical and Chemical Parameters

The evolution of physical and chemical parameters during storage are displayed in Table 2. The initial moisture value of the chocolate filling was 31.7 ± 0.5% *w*/*w*, higher than other traditional formulations [2,4,10] due to the higher cream content used in the present study. At the end of storage time, no significant changes were observed (*p* > 0.01), regardless of the storage temperature or pressure. The initial values of aw were, approximately, 0.92 ± 0.01 (Table 2) higher than other authors [6,10] as a consequence of the higher moisture content [4]. At the end of storage time, no significant changes were observed (*p* > 0.01). The presence of water in a ganache creates a pleasant, creamy smooth and light structure; however, it increases aw and considerably decreases the microbiological stability of chocolate fillings, limiting the shelf life of traditional filled chocolates. The observed values can be considered normal in such type of formulations and are a key factor for limiting shelf life [4].

The initial pH value was 5.20 ± 0.01 (Table 2), slightly below previous results on comparable products [6], due to the acidic profile of the dark chocolate used in the present study. According to the literature, pH values around 5.5 may hinder the growth of bacteria and fungi, however, its viability may not be affected [20]. Additionally, vegetative cells of bacteria are also more susceptible to pressure at lower pH values [21]. Overall, no significant changes were observed after storage time or due to HPP conditions (*p* > 0.01). Although an apparent maximum peak was observed at the 8th and 10th month of Control, the results are not conclusive.

The initial value of colour parameter L* was 44.3 ± 0.9, similarly to previous studies [22,23], decreasing to values between 37.8 and 39.3 at the end of storage time (*p* < 0.01), while neither the storage temperature nor HPP influenced L* (*p* > 0.01). Previous works on chocolate products also concluded that storage time for one year, at 18–20 °C, presented a significant impact on both the sensory and instrumental measurement of chocolate using colorimetry [24]. The initial value of colour parameter a* was 9.3 ± 0.6 (Table 2), slightly higher than in other studies in chocolate products [23], presenting a significant increase during storage (*p* < 0.01) at lower temperatures, namely −12 °C and 4 °C. At a storage temperature of 20 °C, no significant differences were observed (*p* > 0.01), regardless of the HPP conditions. The initial value of colour parameter b* was 10.8 ± 0.5, decreasing during storage on all HPP-processed samples (*p* < 0.01). At the end of the storage time, processing NoHPP/4 presented no significant differences (*p* > 0.01) when compared with month 0, although lower values were observed at the 6th and 8th month. Processing NoHPP/−12 presented no significant differences during storage time (*p* > 0.01).

The mechanical behaviour during storage of Control and 600/20 processing is presented in Figure 1a,b, respectively. Such mechanical spectra shows a strong structure before HPP treatment, with low dependence on frequency and high values of both moduli G′ and G″. After 12 months of storage time, the structure of fillings submitted to 600/20 processing was weaker, based on the higher frequency dependence and cross-over point at low frequency values of mechanical spectra.

The initial value of G′_1Hz_ was 6061 ± 153 kPa (Table 2), followed by a steep decrease during the first two months to values between 208 kPa and 1201 kPa. After, G′_1Hz_ increased until the 12th month, except in Control and NoHPP/4 where no significant differences were observed (*p* > 0.01). At the end of storage time, the highest G′_1Hz_ values were observed in HPP condition 400/10. Previous studies on milk chocolate [25] and filled chocolates [6] also reported a harder texture of staled chocolates than fresh samples, possibly as a result of a change in the crystallinity of the cocoa butter with time [25]. Additionally, some studies about the impact of HPP on the stability of emulsions have concluded that pressure and time of HPP have the capacity to control the lipid crystallization process and change the crystal structure in emulsions [26], which may affect the stability of fat networking created in the ganache [3].

### 3.2. Microbiological Parameters

#### 3.2.1. Total Mesophilic Count

The average values of total mesophilic counts of untreated and HPP-treated chocolate fillings stored for 12 months are presented in Table 3. The initial count of untreated samples was around 3.0 ± 1.9 log cfu/g, similar to previous works on chocolates filled with ganache [10] and some chocolates with fat-based fillings [8]. After HPP processing, counts ranged between 2.5 ± 0.2 and 3.0 ± 0.0 log cfu/g, however, no multiple comparisons were performed since sample sizes did not fulfil the required conditions.

During the first two months of storage time, all samples presented microbiological proliferation (*p* < 0.01), however, at different rates, as a consequence of HPP conditions and storage temperature. From month 2 to month 12, three distinct groups could be identified (*p* < 0.01): (i) NoHPP/−12 and 600/20; (ii) 400/20 and 600/10; and (iii) NoHPP/4, Control and 400/10 (data not shown). As expected, the Control sample presented the highest mesophilic counts during storage, wherein values around 5.4 ± 4.7 log cfu/g could be observed after 12 months of storage (*p* < 0.01), as a consequence of the high aw and high storage temperature [4]. However, from the 2nd month onwards, all conditions identified as group (ii) and group (iii) presented values above the recommended guide values for processed food products [27] and chocolate-based products [20], thus not being acceptable for consumption. On the other hand, both NoHPP/−12 and 600/20 were still acceptable after storage time [20,27], presenting values around 3.8 ± 2.2 log cfu/g (*p* < 0.01). Indeed, the impact of pressure on mesophilic growth during storage has been previously reported in several food products [28,29]. Despite the characteristics of chocolate, namely the low aw, low pH or high proportion of fats and sugar that may prevent an overall microbial growth, the viability of possible microorganisms present is not affected [20]. Particularly, chocolate-based sweets containing fillings with an intermediate aw (0.7 to 0.9) are quite susceptible to microbial growth [20], and some authors refer to problems related to the conservation of chocolate pralines [8].

#### 3.2.2. Moulds

After processing, mould counts were below detection level (Table 3), alike previous studies on reduced-fat white chocolate filling [6]. As expected, HPP presented different impacts over the load of moulds during storage time (Table 3), as already reported in previous works [10]. After 4 months, samples 400/10 presented 1.4 ± 1.2 log cfu/g count, similarly to NoHPP/−12. In both cases, no significant changes (*p* > 0.01) were observed until the end of storage time. The exposure time also presented a considerable influence on the development of moulds, where samples 400/20 presented the first contaminations only after 8 months of storage, with 0.7 ± 0.8 log cfu/g counts, lower than 400/10 processing (Table 3). Such development of moulds during storage time, and after HPP, has been reported previously [28] and may be due to sublethal lesions and a series of intracellular events triggered by HPP, not all of them necessarily lethal [30], but where cell membrane damage has been identified as an important cause of cell death during high pressure processing [31]. Protein denaturation and changes in the active centres of enzymes have also been observed together with changes in enzyme-mediated genetic mechanisms such as replication and transcription, although DNA itself is highly stable to HPP [32]. Indeed, HPP affects non-covalent bond such as hydrogen bridges, van der Wall bonds, etc., which are fundamental for the secondary, tertiary and quaternary structures of proteins. As a consequence, proteins, especially those with complex supramolecular structures will be denatured, which will compromise key metabolic functions of microorganisms. When it comes to the proteins in foods, these may also be denatured, which will ultimately lead to structural changes on the food product, by means of texture, colour, etc., albeit a careful optimization of the processing conditions may be accounted to minimize the impacts of HPP in protein-rich foods [33]. The extent of sublethal lesions is proportional to pressure and a more severely damage population takes longer to recover [28]. In addition, the appearance of moulds after 8 months may be related to the presence of fungi ascospores, which are normally of slow development, with a pressure of 400 MPa (and a dwell time of 20 min) being enough to induce ascospore activation, and further mycelium formation, although at very slow rates due to the acidic media [34]. In fact, the application of 600 MPa in the present study (600/10 and 600/20) was effective against moulds, as no moulds were observed during storage time. In fact, similar pressure values were observed in previous studies where *P. roqueforti* was only completely inactivated at 500 MPa [28]. As expected, untreated samples stored at 4 and 20 °C presented the highest counts during storage (*p* < 0.01) due to a combined effect of storage at higher temperatures, high aw and neutral pH [4].

#### 3.2.3. Yeasts

After processing, yeasts presented a 3.1 ± 2.5 log cfu/g count. In fact, chocolate fillings are considered “intermediate moisture food” (IMF) products [20,35], usually regarded as microbiologically stable; however, sweet IMF products are susceptible to spoilage by xerophilic fungi but also osmophilic yeasts [8,20]. Possible sources of such contamination have been identified as exposure to airborne fungal spores, spread throughout the processing environment, and direct introduction of fungal spores via ingredients [35]. The evolution of yeasts during storage time was similar to the observed previously regarding moulds. In fact, processing 400/10 presented a reduction on yeasts from the initial content of 3.1 to 1.9 log cfu/g, after 2 months storage, increasing later until 2.7 ± 2.1 log cfu/g, at the end of storage time. Such evolution may be caused, as mentioned before, to sublethal lesions caused by HPP treatments and the subsequent time to recover [28,30]. The increase in pressure or time in HPP presented a significant impact on yeasts, as reported before [36], where both 400/20 and 600/10 presented steady values, around 1.0–1.1 log cfu/g, from 6 months to 12 months of storage time, lower than 400/10 (*p* < 0.01) where 2.7 ± 2.1 log cfu/g were observed at the end of storage time. Previous studies have demonstrated the possibility of previously injured cells by HPP recovering their metabolic functions during shelf life evaluation studies, and this may be the case for samples processed at 400/10 and 400/20, as well as 600/10 [37]. Processing 600/20 presented values below the detection level during storage time, which is representative of the efficiency of such high pressure and exposure time, as observed in previous studies [38].

#### 3.2.4. Enterobacteriaceae

Initial counts around 2.5 ± 1.4 log in the *Enterobacteriaceae* group were observed after processing (Table 3). The application of 400 MPa or 600 MPa resulted in an important reduction in this group, below the limit of detection, as observed in previous works [39]. From month 2 until the end of storage time, no *Enterobacteriaceae* counts were observed in any condition (Table 3). Usually, the *Enterobacteriaceae* group can suffer from competition with other bacterial groups for nutrients, namely the lactic acid bacteria (LAB). This group is responsible for the production of bacteriocins with antimicrobial activity, which may explain the reduction in *Enterobacteriaceae* during storage [21]. The presence of LAB during the fermentation of cocoa and the presence in chocolate can result from post-process contamination from ingredients, equipment or environment [40].

## 4. Conclusions

The HPP is a non-thermal treatment used successfully in different food products, especially in high quality products, allowing longer storage time periods. The present study was designed to evaluate the impact of HPP on the long-term storage of chocolate fillings, based on physical, chemical and microbiological parameters. The obtained results concluded on the viability of HPP to improve the shelf life of chocolate fillings, however, depending on applied conditions, namely pressure and time. Moisture and aw did not present significant changes during storage, concluding that polyethylene PET-MET/PE material is an effective barrier against water transfer. Additionally, pH did not present significant changes during storage time or due to HPP conditions. However, physical properties of chocolate fillings were affected during storage. Colour parameter L* decreased in all conditions, parameter a* was affected by lower storage temperatures and parameter b* was affected by higher storage temperatures. Rheological behaviour was also affected, as all samples increased G′_1Hz_ during storage, possibly as a result of a change in the crystallinity of the cocoa butter with time. However, fillings submitted to HPP at 600 MPa for 20 min presented a weaker structure after 12 months of storage time. The microbiological evaluation was also influenced by HPP conditions, especially for yeasts and moulds. HPP at 600 MPa for 20 min showed the highest reduction in mesophiles, yeasts and moulds, compared with Control, and was the only HPP condition to fulfil the recommended food safety criteria after storage time, although storage at −12 °C also presented lower mesophile counts. However, when considering energy costs and energy efficiency during processing and storage, HPP may present a higher sustainability than the compression-vapour refrigeration cycle.

## Figures and Tables

**Figure 1 foods-11-00970-f001:**
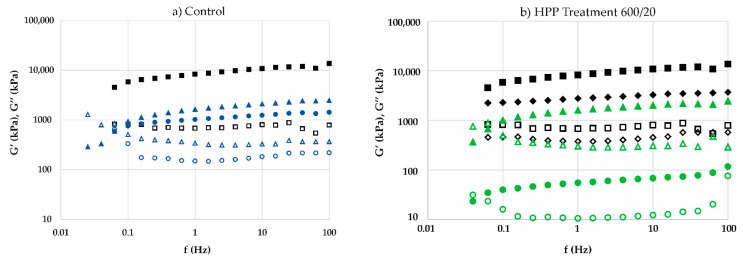
Mechanical spectra of samples: (**a**) Control: after processing (■G′; □G″), after 2 months (●G′; ○G″) and after 12 months (▲G′; ΔG″) of storage; (**b**) Treatment 600/20: after processing (■G′; □G″), after HPP treatment (♦G′; ◊G″), after 2 months (●G′; ○G″) and after 12 months (▲G′; ΔG″) of storage.

**Table 1 foods-11-00970-t001:** Summary of the processing and storage conditions for the chocolate fillings.

Code	HPP Treatment		Storage Temperature (°C)
	Pressure (Mpa)	Time (min)	
Control	0.1	-	20 ± 0.3
NoHPP/−12	0.1	-	−12 ± 2.4
NoHPP/4	0.1	-	4 ± 0.9
400/10	400	10	20 ± 0.3
400/20	400	20	20 ± 0.3
600/10	600	10	20 ± 0.3
600/20	600	20	20 ± 0.3

**Table 2 foods-11-00970-t002:** Impact of different processing and storage conditions on moisture, aw, pH and colour parameters of chocolate fillings. The values are presented as mean ± standard deviation.

Parameter	Months	Control	NoHPP/-12	NoHPP/4	400/10	400/20	600/10	600/20
Moisture % (m/m)	0	31.7 ± 0.5 ^a^	31.7 ± 0.5 ^a^	31.7 ± 0.5 ^a^	31.7 ± 0.5 ^a^	31.7 ± 0.5 ^a^	31.7 ± 0.5 ^a^	31.7 ± 0.5 ^a^
	2	32.5 ± 1.0 ^ab^	32.2 ± 0.7 ^ab^	32.0 ± 0.5 ^a^	33.0 ± 0.6 ^a^	31.9 ± 0.7 ^a^	33.7 ± 0.7 ^ab^	32.0 ± 0.9 ^a^
	4	33.0 ± 0.5 ^ab^	32.7 ± 1.9	32.7 ± 0.6 ^a^	31.9 ± 0.7 ^a^	32.1 ± 0.8 ^a^	32.5 ± 0.9 ^ab^	33.3 ± 0.4 ^ab^
	6	33.8 ± 1.2 ^b^	34.5 ± 1.2 ^ab^	33.2 ± 2.2 ^a^	33.3 ± 2.1 ^a^	34.4 ± 1.0 ^b^	34.3 ± 1.6 ^b^	34.6 ± 1.6 ^b^
	8	31.6 ± 0.5 ^a^	33.9 ± 2.2 ^ab^	33.3 ± 2.1 ^a^	33.1 ± 1.9 ^a^	32.3 ± 0.8 ^a^	33.2 ± 1.4 ^ab^	33.5 ± 1.4 ^ab^
	10	31.8 ± 0.3 ^a^	33.4 ± 0.4 ^b^	33.2 ± 0.6 ^a^	33.0 ± 0.8 ^a^	33.0 ± 0.6 ^ab^	33.5 ± 0.3 ^ab^	33.1 ± 0.6 ^ab^
	12	32.7 ± 0.5 ^ab^	33.2 ± 0.5 ^ab^	32.1 ± 0.2 ^a^	33.8 ±0.5 ^a^	32.0 ± 0.6 ^a^	33.1 ± 0.6 ^ab^	33.1 ± 0.7 ^ab^
aw	0	0.92 ± 0.01 ^ab^	0.92 ± 0.01 ^ab^	0.92 ± 0.01 ^ab^	0.92 ± 0.01 ^ab^	0.92 ± 0.01 ^a^	0.92 ± 0.01 ^ab^	0.92 ± 0.01 ^abc^
	2	0.91 ± 0.03	0.89 ± 0.02	0.90 ± 0.02 ^a^	0.92 ± 0.02	0.91 ± 0.03	0.89 ± 0.04 ^ab^	0.92 ± 0.01 ^a^
	4	0.93 ± 0.00 ^b^	0.87 ± 0.02	0.93 ± 0.00 ^b^	0.91 ± 0.01 ^a^	0.92 ± 0.00 ^a^	0.93 ± 0.00 ^b^	0.93 ± 0.00 ^b^
	6	0.90 ± 0.01 ^a^	0.91 ± 0.01 ^a^	0.91 ± 0.01 ^a^	0.90 ± 0.01 ^a^	0.90 ± 0.03	0.88 ± 0.02 ^a^	0.91 ± 0.01 ^c^
	8	0.93 ± 0.01 ^b^	0.89 ± 0.03 ^ab^	0.93 ± 0.01 ^ab^	0.94 ± 0.01 ^b^	0.94 ± 0.01 ^b^	0.95 ± 0.01 ^b^	0.94 ± 0.03 ^abc^
	10	0.93 ± 0.01 ^ab^	0.94 ± 0.01 ^b^	0.92 ± 0.01 ^ab^	0.93 ± 0.00 ^b^	0.93 ± 0.01 ^b^	0.92 ± 0.01 ^b^	0.94 ± 0.00 ^b^
	12	0.94 ± 0.01 ^b^	0.94 ±0.00 ^b^	0.94 ± 0.01 ^b^	0.93 ± 0.01 ^b^	0.93 ± 0.01 ^ab^	0.93 ± 0.01 ^b^	0.93 ± 0.01 ^ab^
pH	0	5.20 ± 0.01 ^a^	5.20 ± 0.01 ^a^	5.20 ± 0.01 ^a^	5.20 ± 0.01 ^a^	5.20 ± 0.01 ^a^	5.20 ± 0.01 ^a^	5.20 ± 0.01 ^b^
	2	5.14 ± 0.11 ^a^	5.41 ± 0.82	5.12 ± 0.10 ^a^	5.11 ± 0.03	4.94 ± 0.05	4.87 ± 0.08	4.62 ± 0.19 ^a^
	4	5.07 ± 0.07 ^a^	5.13 ± 0.05	5.15 ± 0.07	5.20 ± 0.09 ^a^	5.14 ± 0.05	5.12 ± 0.04	5.16 ± 0.13 ^b^
	6	5.19 ± 0.09 ^a^	5.27 ± 0.05	5.15 ± 0.05	5.22 ± 0.06 ^a^	5.12 ± 0.04	5.13 ± 0.13 ^a^	5.12 ± 0.08 ^b^
	8	5.48 ± 0.19 ^b^	5.58 ± 0.27 ^a^	5.38 ± 0.30 ^a^	5.06 ± 0.09 ^a^	5.06 ± 0.18 ^a^	5.04 ± 0.09 ^a^	5.22 ± 0.15 ^b^
	10	5.70 ± 0.19 ^b^	5.00 ± 0.07 ^a^	5.52 ± 0.15 ^a^	5.08 ± 0.16 ^a^	5.18 ± 0.20 ^a^	5.38 ± 0.16 ^a^	5.22 ± 0.08 ^b^
	12	5.04 ± 0.05	5.40 ± 0.22 ^a^	5.28 ± 0.13 ^a^	5.02 ± 0.04	5.02 ± 0.04	5.38 ± 0.25 ^a^	5.12 ± 0.16 ^b^
L*	0	44.3 ± 0.9 ^de^	44.3 ± 0.9 ^b^	44.3 ± 0.9 ^cde^	44.3 ± 0.9 ^b^	44.3 ± 0.9 ^c^	44.3 ± 0.9 ^c^	44.3 ± 0.9 ^cd^
	2	42.9 ± 0.7 ^bcd^	44.5 ± 1.7 ^b^	45.4 ± 1.0 ^de^	47.2 ± 1.9	48.2 ± 2.3 ^c^	44.9 ± 1.8 ^c^	48.7 ± 0.7 ^e^
	4	43.2 ± 0.4 ^cd^	44.5 ± 1.6 ^b^	43.2 ± 0.5 ^cd^	42.5 ± 1.0 ^b^	43.2 ± 0.8 ^c^	43.1 ± 0.8 ^bc^	42.3 ± 0.7 ^bc^
	6	46.6 ± 1.6 ^e^	44.0 ± 1.8 ^b^	45.8 ± 1.6 ^e^	46.6 ± 1.5 ^b^	45.4 ± 2.2 ^abc^	45.0 ± 1.1 ^c^	47.1 ± 0.9 ^de^
	8	40.4 ± 1.9 ^abc^	38.4 ± 1.8 ^a^	42.4 ± 1.3 ^bc^	40.6 ± 2.3 ^ab^	42.7 ± 1.1 ^bc^	43.0 ± 1.5 ^bc^	40.7 ± 2.6 ^ab^
	10	40.1 ± 0.9 ^ab^	40.0 ± 0.4 ^a^	40.4 ± 0.8 ^ab^	41.0 ± 3.0 ^ab^	40.4 ± 0.8 ^ab^	40.3 ± 1.3 ^ab^	40.5 ± 1.2 ^ab^
	12	39.0 ± 1.0 ^a^	39.3 ± 1.1 ^a^	39.3 ± 0.6 ^a^	38.5 ± 0.6 ^a^	38.4 ± 0.5 ^a^	37.8 ± 1.0 ^a^	38.0 ± 1.1 ^a^
a*	0	9.3 ± 0.6 ^ab^	9.3 ± 0.6 ^a^	9.3 ± 0.6 ^b^	9.3 ± 0.6 ^b^	9.3 ± 0.6 ^ab^	9.3 ± 0.6 ^bcd^	9.3 ± 0.6 ^bc^
	2	8.2 ± 0.6 ^a^	7.6 ± 0.7 ^a^	7.3 ± 0.4 ^a^	6.3 ± 1.3 ^a^	6.5 ± 0.9 ^a^	7.4 ± 0.7 ^a^	6.3 ± 0.3 ^a^
	4	8.3 ± 0.2 ^a^	8.7 ± 0.7 ^a^	9.0 ± 0.3 ^b^	8.5 ± 0.5 ^ab^	8.3 ± 0.3 ^ab^	8.4 ± 0.3 ^ab^	8.8 ± 0.4 ^bc^
	6	8.4 ± 0.3 ^a^	9.0 ± 0.3 ^a^	8.3 ± 0.1 ^ab^	8.5 ± 1.1 ^ab^	8.8 ± 0.5 ^ab^	8.8 ± 0.5 ^bc^	8.7 ± 0.8 ^b^
	8	10.0 ± 0.8 ^b^	11.0 ± 0.4 ^b^	10.3 ± 0.9 ^bc^	9.6 ± 1.1 ^b^	8.9 ± 0.7 ^ab^	9.0 ± 0.5 ^bc^	9.6 ± 1.0 ^bc^
	10	10.2 ± 0.4 ^b^	12.4 ± 0.7 ^b^	12.9 ± 0.4 ^c^	9.8 ± 1.0 ^b^	11.4 ± 1.5 ^b^	10.4 ± 0.7 ^d^	10.7 ± 1.0 ^c^
	12	9.9 ± 0.2 ^b^	11.7 ± 1.1 ^b^	12.0 ± 0.6 ^c^	9.2 ± 0.4 ^b^	10.2 ± 0.9 ^b^	10.1 ± 0.3 ^cd^	10.7 ± 0.9 ^c^
b*	0	10.8 ± 0.5 ^cd^	10.8 ± 0.5 ^a^	10.8 ± 0.5 ^c^	10.8 ± 0.5 ^c^	10.8 ± 0.5 ^b^	10.8 ± 0.5 ^d^	10.8 ± 0.5 ^b^
	2	11.5 ± 0.3 ^d^	10.3 ± 1.0 ^a^	9.7 ± 0.6 ^c^	9.3 ± 1.9 ^bc^	9.9 ± 1.0 ^b^	10.3 ± 1.0 ^d^	9.5 ± 1.1 ^b^
	4	9.5 ± 0.4 ^c^	8.9 ± 1.7 ^a^	10.6 ± 0.4 ^c^	9.5 ± 1.0 ^bc^	9.1 ± 0.7 ^b^	9.4 ± 0.7 ^cd^	10.1 ± 0.7 ^b^
	6	7.6 ± 1.0 ^b^	9.1 ± 0.6 ^a^	8.0 ± 0.6 ^b^	7.1 ± 1.1 ^ab^	8.1 ± 1.3 ^ab^	7.5 ± 1.0 ^bc^	6.6 ± 0.7 ^a^
	8	6.1 ± 0.9 ^a^	9.7 ± 0.4 ^a^	6.2 ± 0.7 ^a^	5.3 ± 1.0 ^a^	4.7 ± 0.6 ^a^	5.4 ± 0.7 ^a^	5.7 ± 0.5 ^a^
	10	6.0 ± 0.6 ^a^	9.4 ± 0.2 ^a^	10.0 ± 0.6 ^c^	6.3 ± 1.0 ^a^	7.2 ± 1.5 ^ab^	6.5 ± 1.1 ^ab^	7.4 ± 1.0 ^a^
	12	5.6 ± 0.4 ^a^	9.9 ± 1.1 ^a^	10.1 ± 0.7 ^c^	5.3 ± 0.5 ^a^	5.8 ± 0.4 ^a^	6.1 ± 0.5 ^ab^	7.2 ± 0.4 ^a^
G′_1Hz_ (kPa)	0	6061 ± 15 ^d^	6061 ± 15 ^c^	6061 ± 15 ^b^	6061 ± 15 ^c^	6061 ± 15 ^d^	6061 ± 15 ^c^	6061 ± 15 ^c^
	2	663 ± 46 ^abc^	419 ± 32 ^a^	1201 ± 36 ^a^	497 ± 3 ^a^	221 ± 17 ^a^	208 ± 18 ^a^	362 ± 40 ^a^
	4	984 ± 23 ^b^	1517 ± 51 ^ab^	551 ± 42 ^a^	1065± 88 ^ab^	692 ± 27 ^ab^	906 ± 25 ^a^	1233 ± 51 ^ab^
	6	1975 ± 98 ^abc^	1600 ± 46 ^ab^	1438 ± 45 ^a^	2087 ± 45 ^b^	2165 ± 49 ^bc^	1984 ± 24 ^b^	1701 ± 19 ^bc^
	8	1899 ± 16 ^c^	1453 ± 12 ^ab^	1720 ± 27 ^a^	1947 ± 71 ^ab^	2579 ± 74 ^abc^	2727 ± 41 ^b^	2250 ± 50 ^b^
	10	1918 ± 43 ^abc^	1576 ± 22 ^b^	1351 ± 73 ^a^	1882 ± 42 ^b^	1908 ± 18 ^c^	2245 ± 22 ^b^	1928 ± 37 ^b^
	12	1985 ± 44 ^abc^	1607 ± 16 ^b^	1541 ± 56 ^a^	2139 ± 40 ^b^	1759 ± 18 ^c^	1767 ± 20 ^b^	1603 ± 19 ^b^

Different superscripts in each column indicate significant difference (*p* < 0.01) during storage period.

**Table 3 foods-11-00970-t003:** Impact of different processing and storage conditions on the evolution of total aerobic mesophiles, moulds, yeasts and *Enterobacteriaceae* during storage time (log_10_ cfu/g). The values are presented as mean ± standard deviation.

Parameter	Months	Control	NoHPP/−12	NoHPP/4	400/10	400/20	600/10	600/20
Total aerobic mesophiles	0	3.0 ± 1.9 ^a^	3.0 ± 1.9 ^a^	3.0 ± 1.9 ^a^	2.9 ± 0.0	3.0 ± 0.0	2.5 ± 0.2	2.9 ± 0.0
	2	5.2 ± 4.3 ^b^	3.9 ± 3.2 ^abc^	5.0 ± 3.8 ^b^	4.9 ± 3.7 ^b^	4.8 ± 4.2 ^b^	4.8 ± 4.0 ^b^	3.9 ± 2.9 ^c^
	4	5.4 ± 4.7 ^b^	3.6 ± 2.5 ^b^	5.2 ± 3.9 ^c^	5.1 ± 4.6 ^ab^	4.9 ± 3.8 ^b^	4.9 ± 4.2 ^bc^	3.6 ± 2.7 ^b^
	6	5.4 ± 4.6 ^b^	3.7 ± 2.9 ^bc^	5.2 ± 3.6 ^cd^	5.2 ± 4.7 ^ab^	5.0 ± 3.8 ^b^	4.9 ± 4.1 ^bc^	3.7 ± 2.7 ^bc^
	8	5.4 ± 4.7 ^b^	3.8 ± 2.7 ^c^	5.2 ± 4.0 ^cd^	5.2 ± 4.6 ^b^	5.0 ± 3.7 ^b^	5.0 ± 3.9 ^bc^	3.7 ± 2.5 ^bc^
	10	5.4 ± 4.7 ^b^	3.8 ± 2.7 ^c^	5.2 ± 4.0 ^cd^	5.2 ± 4.6 ^b^	5.0 ± 3.8 ^b^	5.0 ± 3.8 ^bc^	3.7 ± 2.4 ^bc^
	12	5.4 ± 4.7 ^b^	3.8 ± 2.7 ^c^	5.3 ± 3.9 ^d^	5.2 ± 4.6 ^b^	5.0 ± 4.2 ^b^	5.0 ± 3.7 ^c^	3.8 ± 2.2 ^c^
Moulds	0	<LoD	<LoD	<LoD	<LoD	<LoD	<LoD	<LoD
	2	<LoD	< LoD	2.0 ± 0.9 ^a^	<LoD	<LoD	<LoD	<LoD
	4	<LoD	1.4 ± 1.3 ^a^	2.1 ± 1.4	1.4 ± 1.2 ^a^	<LoD	<LoD	<LoD
	6	2.1 ± 2.1 ^a^	1.3 ± 0.9 ^a^	2.1 ± 1.4	1.4 ± 1.1 ^a^	<LoD	<LoD	<LoD
	8	2.5 ± 2.5 ^a^	1.3 ± 1.0 ^a^	2.0 ± 2.2 ^a^	1.4 ± 1.3 ^a^	0.7 ± 0.8 ^a^	<LoD	<LoD
	10	2.5 ± 2.5 ^a^	1.3 ± 1.0 ^a^	2.1 ± 2.1 ^a^	1.4 ± 1.3 ^a^	0.7 ± 0.8 ^a^	<LoD	<LoD
	12	2.6 ± 2.5 ^a^	1.3 ± 1.0 ^a^	2.3 ± 2.2	1.5 ± 1.4 ^a^	0.7 ± 0.8 ^a^	<LoD	<LoD
Yeasts	0	3.1 ± 2.5 ^a^	3.1 ± 2.5 ^abcd^	3.1 ± 2.5 ^a^	< LoD	< LoD	<LoD	<LoD
	2	4.0 ± 2.9 ^b^	2.4 ± 1.3 ^a^	4.5 ± 3.1 ^b^	1.9 ± 0.8 ^a^	< LoD	<LoD	<LoD
	4	4.3 ± 4.0 ^ab^	3.1 ± 2.6 ^abcd^	4.3 ± 4.0 ^ab^	2.4 ± 1.6 ^b^	< LoD	<LoD	<LoD
	6	4.9 ± 4.0 ^d^	2.7 ± 1.3 ^b^	4.3 ± 3.7 ^ab^	2.6 ± 2.2 ^ab^	1.1 ± 1.1 ^a^	1.1 ± 0.4	<LoD
	8	4.9 ± 4.0 ^d^	2.8 ± 1.5 ^bc^	4.3 ± 3.5 ^b^	2.6 ± 2.1 ^ab^	1.0 ± 0.8 ^a^	1.0 ± 0.8 ^a^	<LoD
	10	4.9 ± 3.9 ^d^	2.8 ± 1.4 ^cd^	4.3 ± 3.5 ^b^	2.6 ± 2.2 ^ab^	1.0 ± 0.8 ^a^	1.0 ± 0.8 ^a^	<LoD
	12	4.9 ± 3.9 ^d^	2.9 ± 1.2 ^d^	4.4 ± 3.5 ^b^	2.7 ± 2.1 ^ab^	1.0 ± 0.8 ^a^	1.0 ± 0.8 ^a^	<LoD
*Enterobacteriaceae*	0	2.5 ± 1.4	2.5 ± 1.4	2.5 ± 1.4	<LoD	<LoD	<LoD	<LoD
	2	<LoD	<LoD	<LoD	<LoD	<LoD	<LoD	<LoD
	4	<LoD	<LoD	<LoD	<LoD	<LoD	<LoD	<LoD
	6	<LoD	<LoD	<LoD	<LoD	<LoD	<LoD	<LoD
	8	<LoD	<LoD	<LoD	<LoD	<LoD	<LoD	<LoD
	10	<LoD	<LoD	<LoD	<LoD	<LoD	<LoD	<LoD
	12	<LoD	<LoD	<LoD	<LoD	<LoD	<LoD	<LoD

Different superscripts in each column indicate significant difference (*p* < 0.01) during storage period.

## Data Availability

The data presented in this study are available on request from the corresponding author. The data are not publicly available due to privacy reasons of research participants.

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
