# Peer review of "Effect of High Pressure on the Properties of Chocolate Fillings during Long-Term Storage"

_foods, 2022, doi:10.3390/foods11070970_

Round 1

Reviewer 1 Report

Choclate filling:

Comments:

Line no 21: aw ? first use full form then abbreviate .

Line no 40: there is no need of reference after 3 words.

Line no 51: what special conditions or ingredients for the growth of pathogenic bacteria?

Line no 63-67. Please check these recent papers, you will get some updated data from these papers, special focus on microbial reduction, enzymatic activities as well as sustainability, please check and cite these recent references:

  1. Impact of High-Pressure treatments on Enzyme activity of Fruit Based Beverages: An Overview. International Journal of Food Science and Technology, 57(2), 801-815
  2. . High-pressure Processing for Sustainable Food. Sustainability, 13(24), 13908.
  3. High-pressure treatments for better quality clean label juices and beverages: Overview and advances. LWT-Food Science and Technology, 141(9), 111828.

Line no 80. Correct it as: 2.1. Preparation

Line no 86: manual filling may cause to increase microbial load. 50 g or mL?

Line no 109: describe the method.

Line no 155: physical and chemical analysis is too long. Try to rewrite in synthesize way.

Line no 167: a excellent work on table data .

Line no 172: does low acidic value impact on pathogenic microbes? Elaborate

Line no 220: mesophilic count value should be given for all treatments.

Line no 234: discussion is not in a proper and linked manner.

Line no 249: how protein denaturation linked with HPP? and how it effct final product quality.

Line no 300: there is no discussion of pH increase in discussion.

Line no 309: conclusion and discussion need to be improved.

Other comments:

You need to discuss some latest results regarding HPP impact on Chocolate. Also use the relevant and updated references to discuss and compare with recent results.  

Reviewer 2 Report

Aurthors investigeted the effect of HPP on the physico-chemical and microbial properties of chocolate filling during long term storage.
Today, it is well known that intermeiate moisture food rarelly results in food poisoning. Therefore, this invesitigation would lead to reduce food poisoning and detorioration in choolate fillings. Reiviewer raised some questions and comments, authors would correct texts and/or tables/figures .

Abstract:Parameter b* decreased at storage temperture 20C to 5.6.
In 2.3 storage conditions, pressurized samples were also stored at 20C.
Thus, the lowest values of b* would be 5.3 at 400MPa/10min. 

Materials and methods:As for small amlitutde oscillatory measurements, samples were evaluated at 20C. Temperatures of NoHPP/-12 and NoHPP/4 samples were significantlly increased. Did these temperature changes have no effect on viscoelastic vehaviour of the filling ?

Reference [4] is cited several times and is important reference.
However, it was not easy to access because it was the original book.
I guess that this book would cite some original papers in a chapter, therefore authors would have better cite the original papers istead of the book[4] to access easier for readers.

Fig.1 shows the mechanical spectra of control and pressurized sample at 600 MPa for 20 min. As for cotrol -"after processig" means 0 month/immediately after processing? -closed square-, the value of G' at 1 Hz was around 10000 kPa. Reviewer could not understand clearly the relationship between Fig1 and Table2.
Authors showed the physical and chemical parameters such as L*, a* , b* and G'. Reviewer totally agree with your obtained results, however the effects of changeing values to consummers are not well written. Authors would have to add explanation of the consummer's acceptable levels for these physical and chemical parameters with citing some references.
In Fig1 and text,a spectrum of G'' was shown and texted, however not shown in Table 2. Authors add results of G'' in Table 2.

HPP inactivation of microorganisms in food was well investigated so far.
Yeast is comparativelly pressure sensitive cells, and it is well known that the population of yeast is siginificantlly decreased immediatelly after pressure treatment at 400 MPa or higher for 10 min. In Table 3, results of month 0 showed reasonable decrease in pressurized samples. However, yeasts grew up to detectable level during storage at 20C even after HPP.
Some microorganisms were lethally injured but only a few cells were not. 
Thus, these partlly injured cells were known to recovery after HPP during storage. Some researcher had already reported this recovery from HPP injury, so authors would cite these reference to discussion.

Scientific name should be written Italics.

Round 2

Reviewer 1 Report

Great work.

Reviewer 2 Report

Authors have carefully revised line by line, and added some refereces.Threfore, I suggest  Accept in present form.